# F6-Net: Algorithmic Reasoning through Gated Pathways and Min-Reduction

## Abstract

Neural Algorithmic Reasoning (NAR) is the research area that aims to build artificial neural networks that can mimic *(classical)* algorithms, reproducing intermediary steps from their execution traces. NAR expects to enhance neural network generalization and help to develop more efficient, adaptable, and faster algorithms. This capability makes it a highly promising approach for dynamic systems in unpredictable, real-world environments. Among the existing methods for algorithm reproduction, the Message Passing Neural Network (MPNN) architecture and its variations, such as Triplet-GMPNN, stand out. This paper proposes a novel variant of Triplet-GMPNN, characterized by three key modifications: a streamlined message-passing process, a new gating-type activation mechanism, and the use of a minimum-type function for embedding reduction. To ascertain the individual contribution of each component, a comprehensive ablative analysis was conducted. This study evaluates each architectural modification through the lens of algorithmic alignment. This work advances the understanding of these systems and opens up new design possibilities for future Neural Algorithmic Reasoning architectures.

## 1 Introduction

Neural Algorithmic Reasoning (NAR) is the research field that builds artificial neural networks to mimic *(classical)* algorithms, reproducing their steps from intermediary abstract inputs (traces) (Veličković & Blundell, 2021; Georgiev et al., 2023; Bevilacqua et al., 2023; Mahdavi et al., 2023; Cappart et al., 2023; Rodionov & Prokhorenkova, 2024). The research area emerged to help neural network generalization enhancement, developing more efficient and faster algorithms, especially those related to algorithmic representations (Mahdavi et al., 2023; Veličković & Blundell, 2021).

Learning algorithms must seem inappropriate at first, once classical deterministic algorithms use a precise, finite, and effective approach so that a set of instructions is capable of transforming an input into an output (Cormen et al., 2022; Arora & Barak, 2009; Knuth, 1997; Robert Sedgewick, 2011). Nevertheless, traditional metrics like output accuracy and correctness lose their conventional meaning when the input data is imprecise. Using a precise algorithm on an imprecise input will not necessarily produce an optimal value.

Neural Algorithmic Reasoning (NAR) is particularly promising for dynamic systems, especially those in real-world, uncontrolled environments. It excels in these contexts because they contain imprecise and noisy metrics, subject to interactive updates (feedback loops) that continuously alter the system's state. Real-time routing problems are an example of a practical application suitable for NAR methods. In such problems, input data, such as distance, speed, and traffic conditions, are constantly changing and imprecise, often forcing conventional algorithms to rely on simplified averages (Rahemi & Mosavi, 2021; Sirish Kumar & Srilatha Indira Dutt, 2020; Rychlicki et al., 2020; Merry & Bettinger, 2019; Zhang & Fink, 2024).

NAR aims to produce models that can generalize beyond training data, providing a robust out-of-distribution (OOD) abstraction, representing algorithm behaviors (Georgiev et al., 2023; Reed & de Freitas, 2016; Zhou et al., 2022). Beyond processing data akin to the training distribution (in-distribution - ID), the models should demonstrate sufficient generalization to larger input sizes and unseen data distributions, ensuring data abstraction (Georgiev et al., 2023; Veličković et al., 2020;

Yan et al., 2020). Provide OOD aims to enhance neural networks' generalization capability while reducing their dependence on large training datasets.

This research presents new method to reproduce algorithms. It is an improvement to the well-known Triplet-GMPNN algorithm (Ibarz et al., 2022). The main contributions of this paper are *(i)* a simplified message passing mechanism for MPNN architectures that operates on node, edge, and graph pair; *(ii)* the usage of a minimum function for effective dimensionality reduction within the activation mechanism; *(iii)* a new gating-type activation mechanism that improved our model performance; *(iv)* a detailed ablation study evaluating the individual impact of each architectural change, allowing for an analysis through the lens of algorithmic alignment.

This work is organized as follows. Section 2 contains a brief background to understand this paper, while related works are presented in Section 3. Section 4 outlines our method. The analysis of experimental results is presented in Section 5. Finally, Section 6 draws some conclusions and proposes future work.

## 2 BACKGROUND

Neural Algorithmic Reasoning (NAR) aims to design neural network models to emulate the execution of algorithms with robust performance on novel and unseen training data distribution (OOD - *out of distribution*), solving instances with higher complexity than those used in the learning phase (Veličković & Blundell, 2021; Xu et al., 2021; Numeroso et al., 2023).

In NAR, an algorithm $\tau$ should be learned by a neural network $\phi$. Considering that the algorithm $\tau$ admits the existence of an abstracted input $\overline{x}$, after the execution, $\tau$ produces an output $\overline{y}$, summarized by $\overline{y} = \tau(\overline{x})$. NAR model expects that $\phi$ receives an abstracted input $\overline{x}$, producing an output defined by $\overline{y} \approx \phi(\overline{x})$ (Veličković & Blundell, 2021; Ibarz et al., 2022; Numeroso, 2024).

Considering that NAR is suitable for imprecise or noisy input data, it is possible to consider that the abstracted input data $\overline{x} \approx x$, where $x$ is the real non-abstracted input. Then, the output $y$ for the real problem is modeled as $y \approx \tau(\overline{x})$, which can be modeled as $y \approx \phi(\overline{x})$, once the differentiable neural network can supposedly adjust the inherited modeled input error.

In NAR, a common paradigm to mimic algorithms is the *encode-process-decode* (Hamrick et al., 2018). In that, $\phi$ represents an encoder-process-decoder neural network composed of an encoder function $\overline{f}$, a processor $\rho$, and a decoder function $\overline{g}$. The processor $\rho$ operates within a latent space, learning a differentiable function that algorithmically maps an input $\overline{x}$ to an output $\overline{y}$.

The simplicity of mapping and reproducing the structures, components, procedures, and calculations of an algorithm $\tau$ into the neural network $\phi$ is defined as algorithm alignment. It means that the model $\phi$ can learn "easily" the algorithm $\tau$ (Xhonneux et al., 2021; Xu et al., 2020; Dudzik & Veličković, 2022). Graph Neural Networks and Transformer architectures stand out in the literature for representing the processor $\rho$. They can converge and extrapolate nonlinear tasks in NAR, representing features of unseen data (Xu et al., 2020; 2021).

## 3 RELATED WORK

Before the rise of Graph Neural Networks (GNNs), Reinforcement Learning (RL) stood out as a differentiable architecture to reproduce algorithms. Even before the definition of Neural Algorithm Reasoning, some authors (Vinyals et al., 2015; Zaremba & Sutskever, 2015; Madsen & Johansen, 2020; Schwarzschild et al., 2021; Graves et al., 2014; Łukasz Kaiser & Sutskever, 2016; Li et al., 2017; Graves et al., 2016; Chang et al., 2019) built methods and architectures to replicate part of the algorithms, as loops, internal states, recursion, and other structures, specially for simple tasks, as addition, multiplication, if loops, and variable assignment.

After 2019, Graph Neural Networks (GNNs) emerged as the prevailing methodology for algorithmic reasoning tasks. This is attributed to their inherent suitability for representing algorithms, especially those based on Dynamic Programming (Xu et al., 2020). The introduction of the CLRS-30 benchmark (Veličković et al., 2022) in 2022 accelerated this trend, creating a standard that biased the use of graph-based architectures. The CLRS-30 framework later inspired new benchmarks, includ-

ing SALSA-CLRS (Minder et al., 2023), which focuses on scalability and sparsity, and CLRS-Text (Markeeva et al., 2024), an adaptation of the original benchmark into a textual format.

Before the establishment of the CLRS-30 benchmark, research in algorithmic learning was characterized by efforts to reproduce algorithms or their core components, including basic structures, if-structures, and loops (Veličković et al., 2020; Veličković et al., 2020; Yan et al., 2020; Xhonneux et al., 2021). A significant limitation, however, was the lack of a standardized evaluation framework. Individual work reproduced a set of algorithms, as Breadth-First Search (BFS), Bellman-Ford, Prim's Minimum Spanning Trees (Veličković et al., 2020), Selection Sort, Merge Sort, and Dijkstra's Shortest Path (Yan et al., 2020). Further applications included developing heuristics for the A* algorithm (Numeroso et al., 2022) and the emulation of the Ford-Fulkerson algorithm, based on max-flow, min-cut theorem (Numeroso et al., 2023).

The introduction of the CLRS-30 (Veličković et al., 2022) dataset provided a benchmark for comparing NAR architectures and accelerated new model development. Ibarz et al. (2022) built Triplet-GMPNN, a generalist learner using triplet messages. Hint-ReLIC (Bevilacqua et al., 2023) added a self-supervision method to improve OOD performance. Triplet Learning was also used as a base method for Triplet Edge Attention (Jung & Ahn, 2023), which included an attention mechanism with edge features, and Triplet-GMPNN with No-Hint (Rodionov & Prokhorenkova, 2023), which removed intermediate supervision, producing a new graph representation without a hint regime.

Different approaches were applied to CLRS-30, as ForgetNet (Bohde et al., 2024), which uses historical embeddings to realign the graph with the algorithms. Memory-enhanced GNNs were presented in Neural Priority Queues (Jain et al., 2023), an extension that uses priority queues as memory. Recursive Algorithmic Reasoning (Jayalath et al., 2023) used a stack as memory for reasoning, tested only in the CLRS-30 DFS dataset, enabling learning recursive structures.

In addition to GNNs, Transformer-based architectures have been used to emulate algorithmic execution. For instance, Li et al. (2024) created an attention mechanism that aggregates information across similar algorithms to improve multitask learning, validating their approach on the CLRS-30 benchmark. Other models, such as Relational Transformers (Diao & Loynd, 2023) and the 2-Weisfeiler-Lehman GNN (Mahdavi et al., 2023), have also been applied to this benchmark. Further advancements include the use of Discrete NAR (Rodionov & Prokhorenkova, 2024) on the SALSA-CLRS dataset to enforce finite-state reasoning, and the Transformer-NAR (Bounsi et al., 2024), a variant of Triplet-GMPNN, applied on both the CLRS-Text and CLRS-30 benchmarks.

Theoretical fundamentals that suggested that neural networks can reproduce algorithms were also presented in the literature. Dudzik & Veličković (2022) established a formal relationship between Graph Neural Networks and Dynamic Programming, suggesting that Message Passing Neural Networks (MPNN) can represent relational structures required for algorithmic execution In related work, Back De Luca & Fountoulakis (2024) demonstrated that Transformer Networks are able to reproduce algorithms and their intermediate steps. Complementing these findings, Georgiev et al. (2024) related equilibrium models and algorithmic processes.

Researchers have also investigated training methodologies for these models. Mahdavi et al. (2023) and Rodionov & Prokhorenkova (2023) analysed the impact of the usage of hints (intermediate supervision) in training, suggesting that practice can lead to lower performance. However, offering a different perspective, Bevilacqua et al. (2023) observed that hints are advantageous when inputs share computational steps (different inputs can have the same intermediate computations), enabling data augmentation, which helps generalization. Xhonneux et al. (2021), in another line of research, studied transfer learning on NAR, indicating that similar algorithms can be used to create an effective bias to help the training of other algorithms.

## 4 METHOD

This section outlines the development and evaluation process for our proposal. Subsection 4.1 describes the CLRS-30 benchmark, used as the basis for our experiments. Subsection 4.2 details the Triplet-GMPNN method (Ibarz et al., 2022), the model on which our work is based. Finally, Subsection 4.3 contains our method, detailing its architecture, components, and our training procedure.

## 4.1 Dataset

The CLRS Algorithmic Reasoning Benchmark (Veličković et al., 2022) features 30 algorithms from the textbook "Introduction to Algorithms" by Cormen, Leiserson, Rivest, and Stein (Cormen et al., 2009), categorized into sorting, searching, dynamic programming, graphs, strings, and geometry. In CLRS-30, all algorithms are modeled as graph problems, regardless of their original domain. Each task's data includes an input, an output, and a series of "hints" - intermediate steps that capture the algorithm's state, enabling a step-by-step replication of its execution. These hints enable tracing the algorithm's execution (Veličković et al., 2022; Diao & Loynd, 2023; Jung & Ahn, 2023).

## 4.2 Foundational Method

The model's architecture employs the *encode-process-decode* paradigm (Hamrick et al., 2018), a structure well-documented in prior works (Ibarz et al., 2022; Mahdavi et al., 2023; Bevilacqua et al., 2023; Jung & Ahn, 2023) and suggested by the authors of the CLRS-30 benchmark (Veličković et al., 2022). To maintain consistency and simplify comparison, this section will use a notation similar to Ibarz et al. (2022).

For each algorithm $\tau$, an encoder $f_\tau$ takes the inputs and hints, which are first concatenated into a single multidimensional vector. The input and hint vectors are concatenated, and the resulting tensor is fed into the intermediate layers of the model (*hidden size*, $h$). Node, edge, and graph embeddings have, at a given time $t$, respectively, the dimensions $n \times h$, $n \times n \times h$, and $h$, where $n$ is the number of nodes and $h$ is the number of hidden size. Algorithms trained on the CLRS-30 benchmark can share these embeddings (Ibarz et al., 2022).

The processor used on the Triplet-GMPNN network (Ibarz et al., 2022) is derived from the well-known Message-passing Neural Network (Gilmer et al., 2017). The model combines the features from nodes, $x_i^{(t)}$, edges, $e_{ij}^{(t)}$, the graph, $g^{(t)}$, and an adjacency matrix, $m_i^{(t)}$, representing the hints. The node features $x_i^{(t)}$ are concatenated with the node embeddings of the intermediate layers $h_i^{(t-1)}$ to create a set of features $z^{(t)} = x_i^{(t)} || h_i^{(t-1)}$, for training the model. Subsequently, the input nodes, $z^{(t)}$, edges, $e_{ij}^{(t)}$, and graphs, $g^{(t)}$, are each transformed independently by a fully connected layer.

Once the fully connected layers generate the embeddings, the model combines them. As previously described, the node and graph embeddings differ in size from the edge embeddings. Consequently, they must be expanded to matching dimensions before the combination step can occur. The combination of the embeddings $m_i^{(t)}$ feeds a Multi-layer Perceptron Neural network, which increases stability and enables the model to capture relationships between data. The resulting embedding is multiplied by the hints, followed by a reduction operation using max aggregation. Finally, both the output $z^{(t)}$ and $m_i^{(t)}$ are independently transformed and combined. In the Triplet-GMPNN algorithm, the model passes through a gating function, defined as $g_i^{(t)} = f_g\left(z_i^{(t)}, m_i^{(t)}\right)$.

The decoder $g_\tau$ is responsible for interpreting the results generated by the processor. It decodes three types of embeddings - scalar, categorical, and pointers - in addition to the final graph output. The decoding process for each embedding type involves computing the similarity between the model's prediction and the target value, using a specific criterion (*argmax* or *softmax*) determined by a code parameter (Ibarz et al., 2022; Veličković et al., 2022).

The loss function is computed after the predictions are decoded. The calculation method is tailored to the data type (scalar, categorical, or pointer) for both the primary predictions and, optionally, hints. If used, hint loss is added to the main prediction loss to help the learning process (Ibarz et al., 2022; Veličković et al., 2022).

## 4.3 Method Contribution

Our analysis of the Triplet-GMPNN architecture (Ibarz et al., 2022), whose implementation is available in the CLRS-30 benchmark (Veličković et al., 2022), identified opportunities to improve the algorithm by simplifying certain structures, all while maintaining performance comparable to the original. For comparison purposes, only changes were made to the processor, despite the possibility of improvements to the encoder and decoder, as existing in the literature (Li et al., 2024).

Keeping the same structure as the MPNN and Triplet-GMPNN approaches, the model combines the embeddings of nodes, $x_i^{(t)}$, edges, $e_{ij}^{(t)}$, and the graph, $g^{(t)}$. Different from the base methods, the embedding nodes $g^{(t)}$ were not concatenated to the intermediate inputs $h_i^{(t-1)}$. Subsequently, the node, edge, graph, and hidden embeddings are each transformed by a dedicated, independent fully connected layer.

The model builds four fully connected networks derived from node, edge, graph, and hidden embeddings, respectively. Since these fully connected models have different dimensions, a second step is performed to ensure compatibility. Node, graph, and hidden embeddings are resized to match the dimensions of the edge-derived model, $n \times n \times h$, as described in Section 4.2. In the tensor, a batch input $b$ is still used, forming a tensor of size $b \times n \times n \times h$.

The outputs of the fully connected layers for each embedding type are combined into a single representation $m_{ij}^{(t)}$, through concatenation. Similarly to the foundational method, the combination of the embeddings $m_i^{(t)}$ feeds a Multi-layer Perceptron (MPL) neural network with an ELU ELU activation function (Clevert et al., 2016). The resulting outputs are reduced through the minimum (min) aggregation function. The chosen aggregation function was selected empirically and outperformed common alternatives (e.g., max, average, sum, etc). After the aggregation, the resulting value is defined as $\bar{m}^{(t)} = min\left(m_{ij}^{(t)}\right)$. Prior to aggregation, the baseline model multiplies its feature representation by the hint information.

The output $m_i^{(t)}$, the nodes, $x_i^{(t)}$, and hidden, $h_i^{(t-1)}$ inputs, are independently transformed and, finally, combined in a single representation. To improve training stability and enable dynamic control over information flow, the representation passes through a linear normalization layer to standardize the feature distributions. Subsequently, this normalized output is fed into a gating function, which emphasizes features. Our gating mechanisms, defined as $g_i^{(t)} = f_g\left(x_i^{(t)}, h_i^{(t-1)}, m_i^{(t)}\right)$, differ from the Triplet-GMPNN representation, multiplying values from hidden output in the gating function.

### 4.3.1 REASONING MODEL

Our initial assessment of the triplet reasoning mechanism suggested that the high dimensionality of the embeddings was producing overly complex fully connected layers. An initial attempt was made to simplify models, avoiding the passage of messages in the format $b \times n \times n \times n \times h$, as found in the literature.

After analyzing the algorithm, we initially opted for a simpler approach, using the $b \times n \times n \times h$ model, with only one edge feature embedding, similar to that found in $\bar{m}^{(t)}$. However, we found that reducing the number of triplet embeddings resulted in poorer results than previously observed - a problem caused not by the data format, but by the lack of variability in the information. So, first, we decided to duplicate the node and graph embeddings to increase data variability. This procedure improved the results obtained, leading to further improvements in our model.

The final messaging model uses doubles of node, graph, and intermediate layer embeddings. To maintain a concise representation and improve variability, edge embeddings were also duplicated. Although increasing variability generally improved results, some empirical unstructured experiments have indicated limited gains from increasing the number of embeddings, leading us to keep a more concise structure for message passing.

## 5 RESULTS

Experiments were conducted using the CLRS-30 benchmark (Veličković et al., 2022), described in Section 4.1. The experiments were developed using the benchmark suite, running in a GeForce RTX 3060 16GB GPU. Some experiments were also run in a Google Colab instance, but those were not reported under time constraints.

We trained each algorithm for 10,000 epochs using a uniform hyperparameter configuration to avoid a biased comparison. The settings applied to all models were a learning rate $lr = 0.001$, a batch size of 16 for all algorithms, and a dropout probability of 0.1. In our experiments, we set the hidden size

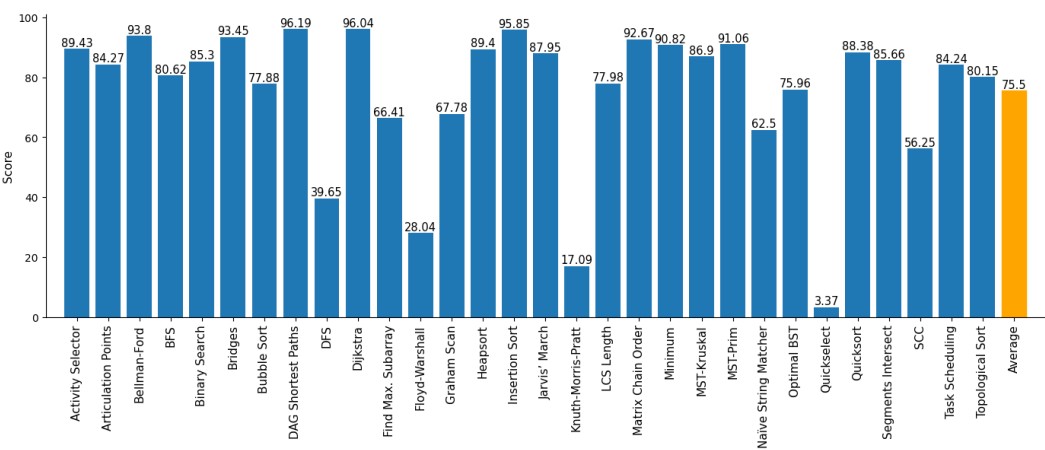

Figure 1: Test scores of our method on the CLRS-30 benchmark.

for node, edge, and graph embeddings to $h = 256$ - variations of these parametrizations are detailed in Section 5.1. No individual parameter tuning was performed. The variations for our experiments are described in Section 5.1.

As illustrated in Figure 1, the algorithms achieved an average accuracy of 75.50%, with the majority (22 out of 30) achieving accuracy above 75%. Table 1 compares our main results against other methods from the literature. For a complete comparison of all algorithms against the literature, see the Appendix A.

Table 1 shows that our method achieved results comparable to the Triplet-GMPNN algorithm (Ibarz et al., 2022) - our reference for architectural simplification. On average, our method achieved a score of 75.50%, performing within a very narrow margin of the Triplet-GMPNN algorithm's 75.98%, with superior performance on multiple algorithms. Furthermore, our algorithm is competitive with existing methods, outperforming several models, despite its simplicity.

Our method performed well, especially on sorting algorithms like Bubble sort, Heapsort, Insertion sort, and Quicksort. In these algorithms, our solution surpassed most of the selected methods in the literature, including the state of the art, Open-Book NAR (Li et al., 2024). This suggests it has characteristics suitable for this type of problem, which warrants further analysis.

Due to the uniform parameterization adopted in the experiments, inferior results were obtained in some algorithms. The method's final average was brought down by poor performance on Depth-First Search, Floyd-Warshall, Knuth-Morris-Pratt, and Quickselect, as available in Figure 1. Furthermore, one of the most unexpected findings was the performance of the BFS algorithm. While it consistently nears 100% accuracy in the literature, it only achieved 80.62% in our experiments. This easily demonstrates that parameter tuning could further improve the results obtained by our algorithm.

## 5.1 ABLATIVE ANALYSIS

The ablative analysis sought to evaluate the impact of specific parameters on the results obtained. Table 2 details the experimental variations for the main algorithms, including changes to the hidden size, activation function, and the removal of the gating activation mechanism.

During initial experiments, the hidden size $h = 128$ was used. However, after setting the parameterization to $h = 256$, an improvement in the average results was observed, making it our preferred setting. However, this choice involved a trade-off. While the average score for $h = 256$ was higher, the median for $h = 128$ is superior. This discrepancy suggests that low results present a greater influence of the average of the algorithms.

For practical applications, the highest-performing results are the most critical, as they demonstrate the method's feasibility. In contrast, average or low scores represent unsuccessful outcomes with high error rates and can be disregarded. Based on the median score, the $h = 64$ setup was optimal,

Table 1: Comparison of our main results with literature methods on the CLRS-30 benchmark. See Appendix A for all algorithm results.

| ALGORITHM | NEURAL NETWORK ARCHITECTURE | | | | | | | | |
|---|---|---|---|---|---|---|---|---|---|
| | A | B | C | D | E | F | G | H | I |
| Bellman-Ford | 97.39 | 96.08 | 98.96 | 99.18 | 87.50 | 94.24 | 96.00 | 95.18 | 93.80 |
| Binary Search | 77.58 | 85.31 | 66.49 | 85.96 | 28.79 | 81.48 | 64.71 | 87.44 | 85.30 |
| Bridges | 93.99 | 97.53 | 84.10 | 99.43 | 1.83 | 37.88 | 72.22 | 99.26 | 93.45 |
| Bubble Sort | 67.68 | 74.35 | 55.51 | 83.19 | 17.78 | 38.22 | 95.78 | 73.16 | 77.88 |
| DAG Shortest Paths | 98.19 | 98.13 | 82.56 | 99.37 | 96.19 | 96.61 | 96.40 | 97.79 | 96.19 |
| Dijkstra | 96.05 | 93.14 | 96.02 | 99.14 | 83.94 | 91.20 | 95.04 | 98.29 | 96.04 |
| Heapsort | 31.04 | 54.42 | 16.28 | 57.47 | 16.37 | 32.96 | 93.07 | 85.71 | 89.40 |
| Insertion Sort | 78.14 | 90.73 | 61.69 | 98.40 | 8.16 | 89.43 | 93.00 | 92.61 | 95.85 |
| LCS Length | 80.51 | 85.67 | 87.75 | 85.43 | 67.50 | 83.32 | 66.91 | 85.54 | 77.98 |
| Matrix Chain Order | 91.68 | 94.46 | 90.45 | 91.08 | 78.74 | 91.89 | 25.12 | 90.85 | 92.67 |
| MST-Prim | 86.39 | 93.30 | 76.68 | 95.19 | 60.31 | 85.77 | 86.60 | 93.41 | 91.06 |
| Optimal BST | 73.77 | 70.70 | 81.82 | 83.58 | 59.89 | 74.40 | 36.04 | 70.04 | 75.96 |
| Quicksort | 64.64 | 55.49 | 46.46 | 73.28 | 17.71 | 39.42 | 94.73 | 83.13 | 88.38 |
| SCC | 43.43 | 44.3 | 32.53 | 53.53 | 22.50 | 28.59 | 48.43 | 65.83 | 56.25 |
| Task Scheduling | 87.25 | 87.70 | 86.34 | 84.55 | 77.79 | 82.93 | 88.08 | 90.93 | 84.24 |

LIST OF ALGORITHMS:

A - Triplet-GMPNN (Ibarz et al., 2022)

B - Triplet Edge Attention Message (Jung & Ahn, 2023)

C - LinearPGN (Mirjanić et al., 2023)

D - ForgetNet / G-ForgetNet (Bohde et al., 2024)

E - Neural Priority Queues (Jain et al., 2023)

F - Relational Transformers (Diao & Loynd, 2023)

G - Recurrent NAR (Xu & Veličković, 2024)

H - Open-Book NAR (Li et al., 2024)

I - Our method, 256-MIN-F6.

as detailed in Appendix B. This variation significantly reduces the number of parameters, resulting in a faster algorithm. Details about the training time of the algorithms can be seen in Appendix C.

After the $h = 256$ configuration produced the best average results, we tested an even larger model with $h = 512$, to explore the possibility of further improvement. This attempt, however, was unsuccessful. The $h = 512$ model yielded worse results, suggesting it may have had difficulty learning effectively. Furthermore, this larger configuration was too memory-intensive, causing some experiments to fail due to insufficient resources.

To benchmark our unconventional minimum aggregation function, we ran a comparative experiment using the maximum function, which is a more standard approach in the literature. The maximum function generally led to worse outcomes, especially for algorithms with moderate to low performance, reducing both the average and median results. However, it's noteworthy that for a few specific algorithms, the maximum function was beneficial, enabling them to achieve their best performance compared to all other variations.

Motivated by studies indicating that multitask learning improves overall results (Chang et al., 2019; Xhonneux et al., 2021), we conducted an experiment to evaluate this technique. The model was trained with algorithms grouped into the following classes according to CLRS benchmark: division and conquer, dynamic programming, geometric, graphs, greedy, searching, sorting, and strings. It was possible to observe an improvement in the performance of a few algorithms compared to previous results. However, on average, the results were inferior to those obtained with individual algorithm training.

Table 2: Ablation study for the main results, evaluating the model's components and parameterizations on the CLRS-30 benchmark. See Appendix B for all algorithm results.

| | NEURAL NETWORK ARCHITECTURE | | | | | | |
| ALGORITHM | 256-MIN-F6 | 128-MIN-F6 | 64-MAX-F6 | 512-MIN-F6 | 256-MAX-F6 | MULTITASK-F6 | NO-GATE-F6 |
|---|---|---|---|---|---|---|---|
| Bellman-Ford | 93.80 | 83.25 | 86.08 | 89.65 | 94.48 | 50.63 | 92.38 |
| Binary Search | 85.30 | 83.64 | 87.30 | 18.80 | 84.03 | 65.43 | 80.91 |
| Bridges | 93.45 | 89.18 | 90.75 | 24.60 | 89.40 | 77.87 | 95.57 |
| Bubble Sort | 77.88 | 92.58 | 84.18 | 71.39 | 84.77 | 84.62 | 87.94 |
| DAG Shortest Paths | 96.19 | 95.41 | 94.34 | 71.19 | 96.14 | 71.24 | 95.31 |
| Dijkstra | 96.04 | 95.21 | 92.72 | 67.77 | 94.68 | 71.19 | 95.02 |
| Heapsort | 89.40 | 79.59 | 25.59 | 83.06 | 86.23 | 90.87 | 82.52 |
| Insertion Sort | 95.85 | 93.07 | 86.72 | 74.56 | 90.09 | 89.50 | 89.36 |
| LCS Length | 77.98 | 85.88 | 85.63 | 86.10 | 86.13 | 85.53 | 85.53 |
| Matrix Chain Order | 92.67 | 91.45 | 88.56 | - | 92.55 | 92.46 | 93.33 |
| MST-Prim | 91.06 | 90.38 | 87.26 | 57.03 | 83.74 | 51.03 | 87.26 |
| Optimal BST | 75.96 | 79.72 | 77.12 | - | 70.73 | 69.85 | 78.80 |
| Quicksort | 88.38 | 79.93 | 90.92 | 84.96 | 92.92 | 77.25 | 93.07 |
| SCC | 56.25 | 39.84 | 49.32 | 29.15 | 18.99 | 28.37 | 26.32 |
| Task Scheduling | 84.24 | 83.83 | 85.27 | 81.74 | 83.78 | 82.52 | 84.60 |

DESCRIPTION OF ARCHITECTURES:

256-MIN-F6 - Original own method, with min aggregation and 256 nodes in the hidden input.

128-MIN-F6 - Variation with min aggregation and 128 nodes in the hidden input.

64-MIN-F6 - Variation with min aggregation and 64 nodes in the hidden input.

512-MIN-F6 - Variation with min aggregation and 512 nodes in the hidden input.

256-MAX-F6 - Variation with max aggregation and 256 nodes in the hidden input.

MULTITASK-F6 - Original algorithm, trained using multitask learning.

NO-GATE-F6 - Variation of the original algorithm without gating mechanism.

Finally, we conducted an experiment to evaluate the performance of the gating-activation mechanism compared to the non-activation baseline version. The results show that combining a gating-activation with linear normalization significantly improved the performance of our method in the majority of the algorithms.

# 6 CONCLUSION

This work introduces a variant of the Triplet-GMPNN (Ibarz et al., 2022) algorithm featuring an improved message-passing mechanism for MPNN networks (Gilmer et al., 2017). The proposed mechanism reduces both the dimensionality and the information content of embeddings during message passing. To complement this change, we developed a novel gating-activation function and employed an uncommon minimum-order method for dimensionality reduction. To analyze the influence of each modification, a comprehensive ablation study was conducted, containing parameter variations that influence the algorithm's performance.

Our solution, despite its simplicity, achieves results that are comparable to those in the current literature. An important advantage is that the models with fewer parameters are highly effective for applications demanding a balance between performance and efficiency. Furthermore, these findings contribute to advancing the understanding of Neural Algorithmic Reasoning (NAR) and help to build new solutions.

For future work, we are planning to focus on both practical enhancements and theoretical grounding. We plan to integrate state-of-the-art architectures from recent literature, using different datasets, as SALSA-CLRS (Minder et al., 2023) and CLRS-Text (Markeeva et al., 2024). We also plan to

conduct a more fine-grained analysis to better understand the architectural characteristics from the perspective of algorithmic alignment.

## LLM-ASSISTANCE DECLARATION

This paper was drafted in Portuguese, translated to English with Google Translate, and then polished using the Gemini LLM. We used the prompt "*rewrite <sentence>*" to refine the language. The AI-generated text was manually reviewed and integrated by the authors.

## REPRODUCIBILITY STATEMENT

The source code required to reproduce this work is available in the supplementary material. It has been anonymized and can be executed using the `run.py` script available in the root directory. The code follows the structure of the publicly available CLRS-30 benchmark source code.

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

APPENDIX

# A COMPLETE RESULTS IN CLRS-30

Table 3 presents the complete results of our algorithm compared with the literature. To ensure a fair comparison, we calculated the average performance using only those works that reported results for all algorithms. In cases where a paper presented several method variations, we chose the one that either performed best or was tested on a greater number of algorithms.

Table 3: Accuracy of literature algorithms in the CLRS-30 Benchmark.

| | NEURAL NETWORK ARCHITECTURE | | | | | | | | | | | |
|---|---|---|---|---|---|---|---|---|---|---|---|---|
| ALGORITHM | A | B | C | D | E | F | G | H | I | J | K | L |
| Activity Selector | 95.18 | - | 95.90 | 90.50 | - | **99.03** | 83.36 | 87.72 | 89.94 | 95.23 | 95.86 | 89.43 |
| Articulation Points | 88.32 | **98.45** | 86.89 | 59.99 | 94.55 | 97.97 | 13.40 | 34.15 | 72.46 | 26.32 | 98.30 | 84.27 |
| Bellman-Ford | 97.39 | 95.54 | 96.08 | 98.96 | 95.25 | **99.18** | 87.50 | 94.24 | 98.58 | 96.00 | 95.18 | 93.80 |
| BFS | 99.73 | 99.00 | 99.85 | 99.50 | 99.95 | 99.96 | 99.64 | 99.14 | **100.00** | 100.00 | 99.99 | 80.62 |
| Binary Search | 77.58 | 89.68 | 85.31 | 66.49 | 85.29 | 85.96 | 28.79 | 81.48 | **94.94** | 64.71 | 87.44 | 85.30 |
| Bridges | 93.99 | 99.32 | 97.53 | 84.10 | 98.91 | **99.43** | 1.83 | 37.88 | 77.96 | 72.22 | 99.26 | 93.45 |
| Bubble Sort | 67.68 | 92.94 | 74.35 | 55.51 | 90.67 | 83.19 | 17.78 | 38.22 | - | **95.78** | 73.16 | 77.88 |
| DAG Shortest Paths | 98.19 | 98.17 | 98.13 | 82.56 | - | **99.37** | 96.19 | 96.61 | 96.60 | 96.40 | 97.79 | 96.19 |
| DFS | 47.79 | - | 59.31 | 20.65 | 34.52 | 74.31 | 6.03 | 39.23 | 37.50 | **100.00** | 40.79 | 39.65 |
| Dijkstra | 96.05 | 97.74 | 93.14 | 96.02 | 96.97 | **99.14** | 83.94 | 91.20 | 95.04 | 95.04 | 98.29 | 96.04 |
| Find Max. Subarray | 76.36 | - | 69.79 | 52.77 | - | 78.97 | 19.94 | 66.52 | 74.30 | **83.53** | 74.52 | 66.41 |
| Floyd-Warshall | 48.52 | **72.23** | 35.33 | 32.52 | 42.16 | 56.32 | 17.84 | 31.59 | 30.32 | 27.49 | 51.52 | 28.04 |
| Graham Scan | 93.62 | - | 94.12 | 87.21 | - | **97.67** | 61.95 | 74.15 | 92.27 | 76.20 | 96.19 | 67.78 |
| Heapsort | 31.04 | **95.16** | 54.42 | 16.28 | 90.67 | 57.47 | 16.37 | 32.96 | - | 93.07 | 85.71 | 89.40 |
| Insertion Sort | 78.14 | 92.70 | 90.73 | 61.69 | 90.67 | **98.40** | 8.16 | 89.43 | - | 93.00 | 92.61 | 95.85 |
| Jarvis' March | 91.01 | - | 93.02 | 81.57 | - | 88.53 | 92.88 | 94.57 | - | 91.83 | **94.74** | 87.95 |
| Knuth-Morris-Pratt | 19.51 | - | 62.59 | 20.59 | - | 12.45 | 3.23 | 0.03 | - | 4.54 | **71.24** | 17.09 |
| LCS Length | 80.51 | - | 85.67 | 87.75 | - | 85.43 | 67.50 | 83.32 | **88.40** | 66.91 | 85.54 | 77.98 |
| Matrix Chain Order | 91.68 | - | **94.46** | 90.45 | - | 91.08 | 78.74 | 91.89 | 91.59 | 25.12 | 90.85 | 92.67 |
| Minimum | 97.78 | 99.37 | 99.23 | 96.08 | **99.98** | 99.26 | 75.65 | 95.28 | 99.43 | 96.92 | 98.65 | 90.82 |
| MST-Kruskal | 89.8 | **96.01** | 78.47 | 85.49 | 92.89 | 91.25 | 43.24 | 64.91 | 87.56 | 67.29 | 91.26 | 86.90 |
| MST-Prim | 86.39 | 87.97 | 93.30 | 76.68 | 85.70 | **95.19** | 60.31 | 85.77 | 76.64 | 86.60 | 93.41 | 91.06 |
| Naïve String Matcher | 78.67 | - | **99.90** | 10.16 | - | 97.02 | 3.40 | 65.01 | 14.08 | 93.71 | 73.57 | 62.50 |
| Optimal BST | 73.77 | - | 70.70 | 81.82 | - | **83.58** | 59.89 | 74.40 | 72.49 | 36.04 | 70.04 | 75.96 |
| Quickselect | 0.47 | - | 3.84 | 1.62 | - | 6.30 | 0.00 | 19.18 | 0.15 | **87.08** | 3.37 | 3.37 |
| Quicksort | 64.64 | 93.30 | 55.49 | 46.46 | 90.67 | 73.28 | 17.71 | 39.42 | 7.41 | **94.73** | 83.13 | 88.38 |
| Segments Intersect | 97.64 | - | 94.96 | 98.81 | - | **99.06** | 93.19 | 84.94 | 96.17 | 97.30 | 98.71 | 85.66 |
| SCC | 43.43 | **76.79** | 44.3 | 32.53 | 66.38 | 53.53 | 22.50 | 28.59 | 50.65 | 48.43 | 65.83 | 56.25 |
| Task Scheduling | 87.25 | - | 87.70 | 86.34 | - | 84.55 | 77.79 | 82.93 | 83.92 | 88.08 | **90.93** | 84.24 |
| Topological Sort | 87.27 | 96.59 | **100.00** | 64.27 | 94.58 | 99.92 | 50.98 | 80.62 | 83.18 | 74.00 | 92.80 | 80.15 |
| AVERAGE | 75.98 | - | 79.82 | 65.51 | - | 82.89 | 46.32 | 66.18 | - | 75.78 | **82.91** | 75.50 |

LIST OF ALGORITHMS:

A - Triplet-GMPNN (Ibarz et al., 2022)

B - Hint-ReLIC (Bevilacqua et al., 2023)

C - Triplet Edge Attention Message (TEAM) (Jung & Ahn, 2023)

D - LinearPGN (Mirjanić et al., 2023)

E - Triplet-GMPNN-NH (Rodionov & Prokhorenkova, 2023)

F - ForgetNet / G-ForgetNet (Bohde et al., 2024)

G - Neural Priority Queues (NPQ) (Jain et al., 2023)

H - Relational Transformers (RT) (Diao & Loynd, 2023)

I - 2-Weisfeiler-Lehman GNN (2WL-GNN) (Mahdavi et al., 2023)

J - Recurrent NAR (RNAR) (Xu & Veličković, 2024)

K - Open-Book NAR (OB-NAR) (Li et al., 2024)

L - Our method, 256-MIN-F6.

## B    COMPLETE ABLATIVE ANALYSIS

Table 4 presents the complete ablation analysis of our method. It details the experimental variations, including changes to hidden size, activation function, and the removal of the gating activation mechanism, applied to all algorithms in the CLRS-30 benchmark (Veličković et al., 2022).

Table 4: Ablation study of the model's components and parameterizations to evaluate their impact on performance on the CLRS-30 benchmark.

| | NEURAL NETWORK ARCHITECTURE | | | | | | |
|---|---|---|---|---|---|---|---|
| ALGORITHM | 256-MIN-F6 | 128-MIN-F6 | 64-MAX-F6 | 512-MIN-F6 | 256-MAX-F6 | MULTITASK-F6 | NO-GATE-F6 |
| Activity Selector | 89.43 | 84.94 | 82.14 | 82.85 | 84.17 | 83.64 | **90.81** |
| Articulation Points | 84.27 | 87.92 | 87.62 | 10.71 | **93.35** | 14.22 | 82.91 |
| Bellman-Ford | 93.80 | 83.25 | 86.08 | 89.65 | **94.48** | 50.63 | 92.38 |
| BFS | 80.62 | 89.36 | **98.68** | 87.21 | 68.85 | 71.39 | 48.00 |
| Binary Search | 85.30 | 83.64 | **87.30** | 18.80 | 84.03 | 65.43 | 80.91 |
| Bridges | 93.45 | 89.18 | 90.75 | 24.60 | 89.40 | 77.87 | **95.57** |
| Bubble Sort | 77.88 | **92.58** | 84.18 | 71.39 | 84.77 | 84.62 | 87.94 |
| DAG Shortest Paths | **96.19** | 95.41 | 94.34 | 71.19 | 96.14 | 71.24 | 95.31 |
| DFS | **39.65** | 15.48 | 21.97 | 32.67 | 18.99 | 29.15 | 29.74 |
| Dijkstra | **96.04** | 95.21 | 92.72 | 67.77 | 94.68 | 71.19 | 95.02 |
| Find Max. Subarray | **66.41** | 64.45 | 47.56 | 59.47 | 61.47 | 66.85 | 64.99 |
| Floyd-Warshall | 28.04 | 21.92 | 24.38 | - | **36.76** | 11.53 | 24.52 |
| Graham Scan | 67.78 | **92.61** | 91.50 | 43.06 | 70.65 | 33.64 | 54.20 |
| Heapsort | 89.40 | 79.59 | 25.59 | 83.06 | 86.23 | **90.87** | 82.52 |
| Insertion Sort | **95.85** | 93.07 | 86.72 | 74.56 | 90.09 | 89.50 | 89.36 |
| Jarvis' March | 87.95 | **91.50** | 89.03 | 53.32 | 90.19 | 59.37 | 80.20 |
| Knuth-Morris-Pratt | 17.09 | **44.63** | 11.04 | 2.10 | 16.16 | 27.05 | 35.60 |
| LCS Length | 77.98 | 85.88 | 85.63 | 86.10 | **86.13** | 85.53 | 85.53 |
| Matrix Chain Order | 92.67 | 91.45 | 88.56 | - | 92.55 | 92.46 | **93.33** |
| Minimum | 90.82 | **96.48** | 94.68 | 91.41 | 95.95 | 85.21 | 96.29 |
| MST-Kruskal | 86.90 | **90.29** | 86.51 | 73.20 | 87.74 | 62.39 | 79.15 |
| MST-Prim | **91.06** | 90.38 | 87.26 | 57.03 | 83.74 | 51.03 | 87.26 |
| Naïve String Matcher | 62.50 | 12.89 | 2.88 | 02.20 | 9.67 | **73.44** | 1.66 |
| Optimal BST | 75.96 | **79.72** | 77.12 | - | 70.73 | 69.85 | 78.80 |
| Quickselect | 3.37 | 1.81 | 0.00 | 0.59 | 0.20 | **3.42** | 0.83 |
| Quicksort | 88.38 | 79.93 | 90.92 | 84.96 | 92.92 | 77.25 | **93.07** |
| Segments Intersect | **85.66** | 85.35 | 81.52 | - | 84.46 | 83.16 | 85.29 |
| SCC | 56.25 | 39.84 | **49.32** | 29.15 | 18.99 | 28.37 | 26.32 |
| Task Scheduling | 84.24 | 83.83 | **85.27** | 81.74 | 83.78 | 82.52 | 84.60 |
| Topological Sort | 80.15 | 62.16 | 77.93 | 60.08 | 76.90 | 23.27 | **82.10** |
| AVERAGE | **75.50** | 73.49 | 70.31 | - | 71.47 | 60.54 | 70.81 |
| MEDIAN | 84.78 | 85.14 | **85.86** | - | 84.31 | 70.52 | 82.72 |

DESCRIPTION OF ARCHITECTURES:

  256-MIN-F6 - Original own method, with min aggregation and 256 nodes in the hidden input.

  128-MIN-F6 - Variation with min aggregation and 128 nodes in the hidden input.

  64-MIN-F6 - Variation with min aggregation and 64 nodes in the hidden input.

  512-MIN-F6 - Variation with min aggregation and 512 nodes in the hidden input.

  256-MAX-F6 - Variation with max aggregation and 256 nodes in the hidden input.

  MULTITASK-F6 - Original algorithm, trained using multitask learning.

  NO-GATE-F6 - Variation of the original algorithm without gating mechanism.

## C    TRAINING TIME

The total training time for all model algorithms with $h = 256$ was approximately 7 hours and 42 minutes, with individual algorithm times ranging from 5 minutes and 51 seconds to 1 hour, 12 minutes, and 18 seconds. For comparison, the runtimes for the models with $h = 128$ and $h = 64$

were approximately 4 hours and 51 minutes, and 4 hours and 32 minutes, respectively. The variance in training time was significant for the slower algorithms, while it was minimal for most of the faster ones. Figure 2 details the training time for each of the algorithms.

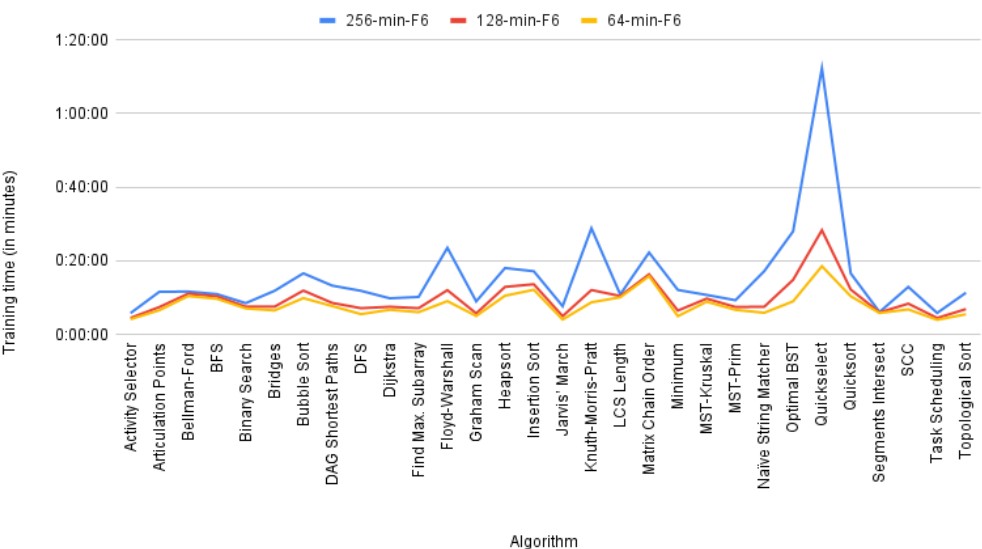

Figure 2: Training time, by method and algorithm.

