# OpenReview forum: "F6-Net: Algorithmic Reasoning through Gated Pathways and Min-Reduction"
_ICLR.cc/2026/Conference — Submitted to ICLR 2026_

### Official Review · Reviewer_hsqW · 2025-10-29

**Soundness:** 1
**Presentation:** 1
**Contribution:** 1
**Rating:** 0
**Confidence:** 5

**Summary:**

The authors propose F6-Net, an extension of Triplet-GMPNN, and evaluate on the CLRS Algorithmic Reasoning Benchmark, where they achieve improved performance on some tasks but do not outperform baseline methods on average.

**Strengths:**

The authors provide some valuable empirical results on modifications to the Triplet-GMPNN architecture that can improve performance on some tasks.

**Weaknesses:**

- The authors contributions are trivial extensions of the Triplet-GMPNN model and do not constitute significant enough research to be published in ICLR
- While F6-Net is better on some algorithms, the proposed model does not improve the average performance across all 30 CLRS-30 algorithms compared to the baseline Triplet-GMPNN. F6-Net also falls short of many other baseline models.
- The provided tables are hard to parse. The authors should bold top performing model and underline the second best model for each tasks. The tables in the main paper should also include the overall average performance.

**Questions:**

See weaknesses section

---

### Official Review · Reviewer_b2JD · 2025-10-31

**Soundness:** 2
**Presentation:** 2
**Contribution:** 2
**Rating:** 2
**Confidence:** 4

**Summary:**

This paper empirically investigates the performance of a graph neural network (GNN) on the DeepMind CLRS-30 benchmark for algorithmic reasoning.  The GNN tested in this paper is a variation on a previously published network, called Triplet-GMPNN, with the following modifications:

- Graph edge embedding vectors are aggregated by a min function, rather than a max or average
- Not concatenating hidden embeddings with node embeddings
- Multiplication by hidden embeddings in the gating function

The main results show that this paper's method is competitive with Triplet-GMPNN and several other prior works, despite using what is claimed to be a simpler architecture.  Additional results compare several different hyper-parameter settings (such as layer sizes) and their effects on performance.

**Strengths:**

- Using a min aggregation function is interesting, and might be interpreted as something more like a logical-and operation whereas max is more like logical-or.  It makes sense that certain algorithmic tasks would benefit more from one vs the other.
- The paper includes experiment code in supplementary material, aiding reproducibility.
- The paper includes a fairly broad evaluation of several hyper-parameter settings and model ablations.

**Weaknesses:**

- The contributions are somewhat incremental: The paper tweaks an existing architecture, rather than proposing a fundamentally new architecture.  It also is not so clear to me that the tweaks are significantly more "simple" than the original architecture as is claimed in the paper.  An aggregation function is changed, and certain vectors are multiplied rather than concatenated.  The latter should make the model's footprint a bit smaller, but I am not sure it is substantially smaller or conceptually simpler.
- The proposed method is comparable to baselines but is by no means a clear winner.  In fact, it appears that overall the baselines perform better: In Table 1, the open book baseline wins in a majority (9 out of 15) algorithms.
- The paper is entirely empirical, without any theoretical arguments or insights into why the changes should perform comparably.  In particular, there is no solid, evidence-based explanation for why a min aggregation function should perform comparably to, or better than, more conventional choices.
- The experiments do not include any error bars, standard deviations, or statistical tests to quantify reproducibility.  It would be better to do multiple independent runs of each experiment and then average the results across the random noise in each run (model initialization, dropout, etc.).  This is particularly important since the modified model and baselines performed very similarly.
- There were a number of issues with the paper's presentation.  I understand the authors are not native English speakers, but some of these issues are independent of language:
    - section 3 refers to RL as a "differentiable architecture" but RL is an architecture-agnostic training paradigm, not an architecture.
    - There is a repeated reference line 113
    - missing period line 142
    - Lines 178-180 contain a redundant mention of concatenation
    - The same symbol "g" is used to denote multiple independent things.  In 4.2 this symbol is used to represent the graph embedding, the output of the gating function, and the decoder.  Different symbols should be used for each.
    - An architectural diagram would have been more helpful than purely textual descriptions, especially for new contributions in 4.3, and there is enough space in the 9-page limit to include one.
    - line 230: "MPL" for multi-layer perceptron is backwards, "ELU" is duplicated twice
    - Section 5 mentions learning rate but not which optimization method was used (SGD? Adam? something else?)
    - Tables 1,2 would be easier to parse with bold-face indications of the best results
    - Some textual descriptions would benefit from more precise formulas.  For example, line 240-242 just writes the gating function as $f(x,h,m)$ without any formula clearly defining $f$ (it is only described verbally).
    - The method is called F6 in the title, but this name is never explained in the main text.

**Questions:**

- Can the authors provide any intuition why min aggregation should work comparably to, or better than, mean/max aggregation?
- In 4.3.1: The "duplication" description was not clear to me.  What is the difference between the two duplicates that increases variability?  Do they have different trainable parameters? Is there any message passing between them or they each independently compute separate predictions?  If it is the latter, how do you combine the predictions from each duplicate at the end?

---

### Official Review · Reviewer_Puca · 2025-11-03

**Soundness:** 3
**Presentation:** 2
**Contribution:** 2
**Rating:** 2
**Confidence:** 3

**Summary:**

The paper builds on the Triplet-GMPNN model, proposing various modification of the core architecture. This modification are evaluated on the CLRS-30 benchmark, the original benchmark used for Triplet-GMPNN, and shows some improvement in performance. In addition to this, the author runt a few ablation to justify the modifications done to the core architecture.

**Strengths:**

The paper is in some sense well structured and follows a typical recipe, which makes it easy to parse. The authors focus on algorithmic reasoning, which is a well defined problem with practical implications. They build on a well-performing architecture in this space, providing some alteration to the architecture that are easy to implement, and which empirically lead to good numerical results. The authors do some ablation of their method as well, showing that the alteration indeed lead to a improved performance.

**Weaknesses:**

While the overall structure of the paper is good, the writing is not great. For some reason, the authors avoid to provide mathematical formulation of what they do, describing the modifications to the backbone mostly in plain english. I think this actually leads to lack of clarity. I would have liked to see the formula of the proposed gating. Or some pseudo-code. I think the way it is expressed in english, is not actually clear to me what the new proposed gating is. The text also does not make any attempts of clearly singling out what is part of the original backbone architecture, versus what is novel. The Triplet-GMPNN architecture is described in section 4.2 (which I think it should since we will be mostly working on it) again in plain text, in a very confusing way. And the "triplet" part of the architecture, one of the main innovation from general message passing networks is not really described in this section, even though the section is almost an entire page. I think is more likely to expect readers of the paper to be knowledgeable about message passing networks but less knowledgeable about the specific architecture you are building on, then to assume they know the architecture but somehow a generic description of how message passing architectures work is needed.

I believe that either a diagram, or some math or some pseudocode would help immensely the paper, making it clear what is the novel alteration of the paper, what is the architecture being run etc,

My second issue is with the scale of results and ablations. From reading the text, I'm not sure there is any deep motivation of the proposed modifications (in a mathematical sense or otherwise), which is fine. ML is to a large degree an empirical field, and if a different formulation of the architecture works better, that should be enough. But when the paper does not have any of these deeper reasoning for changing the gating function or the aggregator to min, I feel more extensive results are needed. And more extensive ablation (and not on the hidden size h -- but rather on the architectural changes being proposed). I think the scale of the experiments, the lack of deeper motivation and the scale of ablation are not sufficient for a venue like ICLR.

**Questions:**

1. Can you explain specifically how the new gating function works? (section 4.3 if I'm not mistaken).
2. Can you clearly describe the novelty of the work. What changes have you done to the backbone architecture?
3. Is there a way to motivate more widely why this changes are significant?

**Details Of Ethics Concerns:**

No reason for ethics review.

---

### Official Review · Reviewer_HV4Q · 2025-11-10

**Soundness:** 4
**Presentation:** 3
**Contribution:** 3
**Rating:** 4
**Confidence:** 3

**Summary:**

This paper proposes an improved version of Triplet-GMPNN, the previous state of the art for Neural Algorithmic Reasoning (NAR). Specifically, it improves message-passing, gating, and aggregation.

**Strengths:**

The specific improvements proposed over the TGMPNN architecture are interesting, and the authors demonstrate a good understanding of the existing literature.

Some performance improvements, namely the heapsort performance, are quite notable.

The authors cover a quite comprehensive series of baselines.

The ablation study covers several variants of the proposed architecture.

**Weaknesses:**

Statistics are not reported in Tables 1 and 2 or Figure 1, making it quite difficult to tell if the changes in performance are because of the architecture, or because of other factors. It's also not fully clear where numbers come from author-run experiments and where they're reproduced from previous research, e.g. are all initializations, hyperparameters and other factors controlled for? There are also several cases where the proposed architecture does not outperform the baseline (which I'd consider to be Triplet-GMPNN), e.g. Bellman-Ford, DAGSP.

The omission of BFS from the main results seems dishonest, it's an important and easy algorithm within NAR that most capable methods can achieve 100% accuracy on. Commenting on this deviation and others would help the paper. However the 64-min variant performing the best on BFS is quite interesting.

There is no significant pattern in the results between the architecture variants tested -- I'm not sure if the takeaway is that these variants should be used for each algorithm, or that one specific variant is a best general architecture?

Related work section first paragraph is rather weak, and does not explicitly mention important previous architectures such as RNN/LSTM, NTM. It may be more important to describe the features and shortcomings of other MPNN architectures and what led to the improvements proposed in the paper. Also, the description of "Simulation of graph algorithms with looped transformers," keep in mind that in may be useful to distinguish between theoretical expressivity and practical trainability, see [1]

The paper is slightly under max-length, and further space could be used to address these concerns. Overall writing and format could be improved to make the paper more publication ready.

[1] Delétang, G., Ruoss, A., Grau-Moya, J., Genewein, T., Wenliang, L. K., Catt, E., ... & Ortega, P. A. (2022). Neural networks and the chomsky hierarchy. arXiv preprint arXiv:2207.02098.

**Questions:**

Why do the authors think sorting algorithms are represented so well by their proposed architecture?

Are the results reported in length-generalization using the default CLRS test-train split?

Which results come from author-run baselines, and which are reported from other papers?

A consolidated reporting of hyperparameters and experiment configurations (at least in the appendix), would be quite helpful (beyond what section 5 paragraph 2 lists).

---

### Meta-Review · Area_Chair_mjBA · 2026-01-06

**Summary:**

Clear rejection due to unanimous low reviewer scores below the acceptance threshold. No rebuttal submitted or other attempts made for authors to address feedback.

**Reviewer Concerns:**

No rebuttal submitted.

**Reviewer Scores:**

No rebuttal submitted.

---

### Decision · Program_Chairs · 2026-01-26

Reject